# Novel Chemical Cross-Linked Ionogel Based on Acrylate Terminated Hyperbranched Polymer with Superior Ionic Conductivity for High Performance Lithium-Ion Batteries

**DOI:** 10.3390/polym11030444

**Published:** 2019-03-07

**Authors:** Kang Zhao, Hongzan Song, Xiaoli Duan, Zihao Wang, Jiahang Liu, Xinwu Ba

**Affiliations:** 1College of Chemistry & Environmental Science, Hebei University, Baoding 071002, China; Kamjue@163.com (K.Z.); 15733290354@163.com (Z.W.); 15175734522@163.com (J.L.); 2School of Chemistry and Materials Science, Ludong University, Yantai 264025, China; xlduan04@iccas.ac.cn

**Keywords:** ionogels, hyperbranched polymers, ionic liquid, chemical cross-linking, lithium-ion batteries

## Abstract

A new family of chemical cross-linked ionogel is successfully synthesized by photopolymerization of hyperbranched aliphatic polyester with acrylate terminal groups in an ionic liquid of 1-butyl-3-methylimidazolium tetrafluoroborate (BMIMBF_4_). The microstructure, viscoelastic behavior, mechanical property thermal stability, and ionic conductivities of the ionogels are investigated systematically. The ionogels exhibit high mechanical strength (up to 1.6 MPa) and high mechanical stability even at temperatures up to 200 °C. It is found to be thermally stable up to 371.3 °C and electrochemically stable above 4.3 V. The obtained ionogels show superior ionic conductivity over a wide temperature range (from 1.2 × 10^−3^ S cm^−1^ at 20 °C up to 5.0 × 10^−2^ S cm^−1^ at 120 °C). Moreover, the Li/LiFePO_4_ batteries based on ionogel electrolyte with LiBF_4_ show a higher specific capacity of 153.1 mAhg^−1^ and retain 98.1% after 100 cycles, exhibiting very stable charge/discharge behavior with good cycle performance. This work provides a new method for fabrication of novel advanced gel polymer electrolytes for applications in lithium-ion batteries.

## 1. Introduction

Recent years have witnessed a great deal of attention on the development of gel polymer electrolytes (GPE) in many various electrochemical devices [1,2,3]. Commonly, GPE is composed of host polymers and conducting salts dissolved in organic solvents [4,5]. However, some disadvantages include low ionic conductivities and narrow electrochemical windows, especially, the flammability and toxicity of the volatile solvents limits its application [6,7,8]. Ionogels, based on ionic liquids with three-dimensional networks, are promising new GPE materials and have received considerable attention owing to their attractive properties such as high ion mobility, high thermal stability, safety, and nonflammability [9,10,11,12,13,14,15]. These characteristics of ionogels allow them to be considered as attractive candidates for many applications in energy storage devices, actuators, and flexible electronics [16,17,18,19,20]. To our knowledge, currently, the investigation of ionogels has mainly focused on the electrolyte based on the linear polymer as a homopolymer/copolymer matrix; however, the high crystallization or glass transition temperature (*T*_g_) makes it difficult to match well with ionic conductivity and modulus [21,22,23]. On the other hand, the networks of noncovalent associations (hydrogen bonding, host-guest interaction, phase separation or crystallization) for many physical cross-linked ionogels can be easily broken and show weak mechanical stability at higher temperatures [24,25,26]. Therefore, it is both desirable and challenging to solve the breakage problem and prepare high-performance chemical cross-linked ionogels from the functional polymer with low crystallinity or *T*_g_.

Highly branched polymers, as one of the fastest growing polymers in function macromolecular materials, have captured more and more attention because of their unique structural characteristics and properties such as low crystallinity, controlled flexibility, and various functionalities [27,28,29,30,31,32,33,34]. Hyperbranched polymers (HPs), a unique class of branched polymer with a large number of terminal groups, have been used as hot hosts for use in ion-conductive polymer electrolytes due to their completely amorphous and low glass transition temperatures [35,36,37,38,39]. Commonly, most hyperbranched polymers contain heteroatoms and the branches show high segmental motion ability, resulting in a relatively high room temperature ionic conductivity [40]. For example, Lee et al. found that poly(ethylene oxide)s with varying degrees of hyperbranching were effective at preventing the crystallization of PEO and led to approximately a 100-fold increase in the Li-ion conductivity below 50 °C as compared with linear PEO. Moreover, a large number of functionalized terminal groups make hyperbranched polymers improve the mechanical properties of the polymer electrolytes by introducing crosslinking [41]. Itoh et al. found that the cross-linked composite polymer electrolytes of cross-linkable hyperbranched polymer capped with acryloyl group showed higher tensile strength than the non-cross-linked composite polymer electrolyte. Importantly, lots of the hyperbranched polymers can be easily dissolved in ionic liquids due to many kinds of polar groups in its side chain, such as acrylate (the hyperbranched star polymer with hyperbranched polystyrene as the core and polymethyl methacrylate block poly(ethylene glycol) methyl ether methacrylate) and amino end groups (hyperbranched polyamidoamine) [42]. Therefore, the unique branched architectures of hyperbranched polymers make it one of the most polymer matrixes for preparing high performance ionogels with high ionic conductivity and high mechanical properties.

In this paper, we synthesized a novel chemical cross-linked ionogel for the first time by photopolymerization of hyperbranched aliphatic polyester with terminal acryloyl groups in ionic liquid BMIMBF_4_. Microstructure, viscoelastic behavior, mechanical properties, thermal stability, and ionic conductivities for these ionogels have been comprehensively studied. Additionally, a Li battery cell was successfully fabricated by using the obtained ionogels as electrolytes and was found to display excellent, highly specific capacity and good cycle stability.

## 2. Experimental Methods

### 2.1. Materials

2-ethyl-2-(hydroxymethyl)-l,3-propanediol (TMP), p-toluene sulfonic acid (PTS), acrylic acid, 2, 2-bis(hydroxymethyl)propionic acid (DMPA), 1-butyl-3-methylimidazolium tetrafluoroborate (BMIMBF_4_) (Ionic liquid), 2-Hydroxy-4′-(2-hydroxyethoxy)-2-methylpropiophenone (photoinitiator), lithium borofluoride (LiBF_4_), *N*,*N*-dimethylformamide (DMF), and poly(vinylidene fluoride) (PVDF, *M*_w_ = 534,000) were purchased from Sigma-Aldrich (Shanghai, China) and used as received.

### 2.2. Synthesis of Acrylate Terminated Hyperbranched Polymers (HP-A)

Acrylate terminated hyperbranched polymers were synthesized via a two-step process (Figure 1) [43]. First, hyperbranched aliphatic polyester (HP-OH) was synthesized via a pseudo-one-step reaction by using 2, 2-bis(hydroxymethyl)propionic acid (DMPA), (9.49 g, 70 mmol), 2-ethyl-2-(hydroxymethyl)-l,3-propanediol (TMP), (1.34 g, 10 mmol) and a catalytic amount of PTS reacted in an oil bath preheated to 140 °C. The mixture was left to react under a stream of nitrogen within 1 h, removing the water formed during the reaction. Then, the reaction continued under a vacuum for an additional 2 h. Then, another 8.05 g (60 mmol) of DMPA was added in the reaction vessel and a vacuum was applied again for 2 h. After the reaction, a white waxy solid product of HP-OH with 16 hydroxyl was obtained (Appendix A). The molecular weight of HP-OH was measured to be 1700 g/mol by GPC measurement. Second, HP-A was synthesized by acrylate modification of HP-OH [44]. Eight grams of HP-OH (80 mmol OH-groups) was added to a round-bottomed flask and preheated until it melted in an oil bath preheated to 120 °C. Then 2.88 g (40 mmol) of acrylic acid, catalytic amounts of the inhibitors hydroquinone and nitrobenzene, and 8 g of toluene were added, and the temperature was set to 110 °C. When a homogenous mixture of the added reagents was obtained, 0.5 g of the catalyst, PTS, was added. The reaction mixture was refluxed for 1 h under constant stirring and air flow to prevent premature cross-linking. The reaction was then quenched by immersing the flask into an ice-water bath. The crude sample was poured into a large amount of water. The precipitate was washed with a large amount of water several times and, after being dried under a vacuum, brown viscous products of HP-A with eight double bonds were obtained. The substitution degree of hydroxyl groups by acrylic groups can be calculated from the ^1^H NMR spectrum; the degree of acrylation is 50% and HP-A has eight double bonds (Appendix A) [45]. The molecular weight of HP-A was measured to be 2200 g/mol by GPC measurement.

### 2.3. Synthesis of Ionogels by UV Curing

A certain quality of HP-A (1.0 g, 1.5 g, 2.0 g, 2.5 g and 3.0 g) was dissolved in ionic liquid of BMIMBF_4_ (9 g, 8.5 g, 8 g, 7.5 g, and 7 g) under magnetic stirring at 60 °C for 2 h to obtain a homogenous mixture with the weight fractions of 10, 15, 20, 25 and 30 wt % of HP-A, respectively. To this HP-A/IL mixture, 1 wt % (mass fraction with respect to HP-A content) of 2-Hydroxy-4′-(2-hydroxyethoxy)-2-methylpropiophenone (as the initiator) was added and stirred for 1 h at 60 °C. The obtained mixers were put into an ultraviolet exposure chamber (SCIENTZ 03-II, Ningbo Scientz Biotechnology (Co., Ltd., Ningbo, China). After being irradiated with a power of 50 W and a wavelength of 365 nm for 2 h, the chemical crosslinking ionogels with 10, 15, 20, 25, and 30 wt % of HP-A were obtained and were named Ionogel-1, Ionogel-2, Ionogel-3, Ionogel-4, and Ionogel-5, respectively. The chemical structures of ionic liquids and acrylate terminated hyperbranched polymer (HP-A) and the reaction scheme are given in Figure 1.

### 2.4. Preparation of the Ionogel Electrolyte Membranes

The ionogels electrolytes membranes (HP/ionic liquid/PVDF/LiBF_4_) were prepared by solution-casting and UV curing techniques. The hyperbranched polymer (HP-A), photoinitiator, lithium salt, ionic liquid, and poly(vinylidene fluoride) (PVDF) were added to *N*,*N*-dimethylformamide (DMF), and then the solution was poured into Teflon mold. After drying, the obtained mixers were irradiated with a power of 50 W and a wavelength of 365 nm for 2 h, and the chemical crosslinking ionogels electrolyte membranes were obtained. In this experiment, the ionogel electrolyte membrane of 30% HP-A/50% BMIMBF_4_/10% PVDF/10% LiBF_4_ was used.

### 2.5. Characterization

FTIR spectra were recorded in the region of 400–4000 cm^−1^ for each sample on a Varian-640 spectrophotometer (Varian, CA, USA). The spectrum for each sample was obtained from averaging 32 scans over the selected wave number range. Rheological measurement were performed on a stress-controlled rheometer (TA-AR2000EX, TA Instruments, Liverpool, UK) equipped with a parallel-plate geometry (diameter 25 mm). Before the oscillatory shear measurements, a strain sweep from 0.1 to 100% with a fixed frequency of 6.28 rad/s was performed for each sample to determine the linear viscoelastic regime. The chosen strains of 1–8% fell well within the linear viscoelastic regime for the frequency range of 0.01–100% rad/s in the oscillatory shear measurement. The experimental temperature was mainly set at 25 °C. The dynamic temperature sweep measurements at an angular frequency of 6.28 rad/s were conducted from 25 to 200 °C with a heating rate of 1 °C/min. For each sweep measurement, repeat specimens were requested, and the number of repeat specimens was three in order to examine the data reproducibility. All measurements were conducted under a nitrogen atmosphere. Ionic conductivities of the ionogels were measured using a Zahner Zennium pro electrochemical workstation (Zahner, Kronach, Germany) over the frequency range from 1 Hz to 500 kHz at an AC oscillation of 10 mV. The measurements of the samples, having a diameter of 8 mm and thickness of 2 mm, were carried out in a cell which consisted of a Teflon spacer sandwiched between two platinum-coated stainless-steel electrodes. The cell constant was determined using a 0.01 M KCl aqueous solution at 25 °C as the reference. Ionic conductivities were calculated from the bulk resistances obtained from the impedance spectra. The minimum in the Nyquist plot of the negative imaginary part of the impedance versus the real part of the impedance was taken as the sample resistance, *R*. The ionic conductivity, σ, was calculated as *d*/(*RS*), where d and S are the thickness and area of the sample, respectively. Scanning electron microscopy (SEM) images of the samples were obtained on a JEOL SEM 6700 operating at 5 kV, and the ionogel samples were first surface treated and then coated with platinum before the SEM observation. ^1^H NMR spectra was obtained from a 800 MHz Bruker Aspect NMR, using DMSO-d6 and CDCl3 as solvents. Thermal analysis of ionogels was performed using a Perkin Elmer TGA 6 (Perkin Elmer Instruments, Fremont, CA, USA) under nitrogen atmosphere. The temperature ranged from 25 to 800 °C with a heating rate of 15 °C/min. For TGA measurement, three specimens were analyzed to assess data reproducibility.

## 3. Results and Discussion

### 3.1. Synthesis and Microstructure of Ionogels

The chemical cross-linked ionogels were synthesized by photopolymerization of acrylate terminated hyperbranched polymer (HP-A) in ionic liquid BMIMBF_4_ (Figure 1). HP-A with eight double bonds was synthesized by acrylate modification of hyperbranched aliphatic polyester HP-OH which has 16 hydroxyl end groups per molecule. Furthermore, the FTIR spectra were used to confirm the polymerization, and the results are illustrated in Appendix A. Note that ionogel samples were treated and clean up the ionic liquid (named as xerogel) to show signals clearly in FTIR experiments. The FTIR spectrums of HP-OH shows the bands of -OH 3500–3100 cm^−1^ and the C=O (ester carbonyl) stretching band at 1721 cm^−1^. For HP-A, the band of -OH at 3500–3100 cm^−1^ displays an obvious decrease as compared to HP-OH and the percentage reduction is about 50%, indicating that half of -OH is converted. However, the characteristic bands of C=C peaks at 1637 cm^−1^ and 1616 cm^−1^ for acrylic groups appear. After polymerization, for the obtained ionogels and xerogel without ionic liquid, the C=C peaks at 1637 cm^−1^ and 1616 cm^−1^ are disappeared, indicating that all of the acrylate terminated hyperbranched polymer has been completely polymerized. More importantly, for the spectrum of ionogels compared with xerogel without ionic liquid, the characteristic absorption peak of C=O shifts to a higher wavenumber, which corresponds to the noncovalent interactions among the polymer matrix and ionic liquids^34^. Moreover, the glass transition temperature (*T*_g_) at −7.5 °C for HP-A can be observed in DSC curves (Appendix A). Note that the obtained ionogels show no *T*_g_ in a wide temperature range from −50 to 100 °C, suggesting that it can be used as a good electrolyte for low-temperature electronic devices.

The samples of the ionic liquid solution before curing exhibit good transmittance and fluidity for molding; therefore, the obtained ionogels have light transmission and are easy to process into a round shape (Figure 2a,b). Moreover, the self-supporting ionogel film can be bent (Figure 2c), which demonstrates that the ionogel has flexibility. To study the microstructure of ionogels in detail, SEM measurements were carried out. Figure 2d,e presents typical SEM images, showing the morphologies of the freeze-dried samples. It can be seen from Figure 2d that the cross section surface is extremely rough and contains lots of holes. In Figure 2e, at a higher magnification, the nanometer-sized pores with sizes about 100 nm are observed. Interestingly, the pore wall has a beads-shaped network structure which is constructed from cross-linked spherule particles of HP-A, and the particle size is about 20 nm in diameter. It is known that the size of the ions of [BMIM]^+^ and [BF_4_]^−^ for ionic liquid BMIMBF_4_ is less than 1 nm [46], which is far smaller than the size of the hole and interior cavities of hyperbranched polymer, making it possible for ions to move freely and quickly, and, therefore, the ionogels will possess superior ionic conductivity.

### 3.2. Viscoelastic and Mechanical Properties of Ionogels

The dynamic viscoelasticity of ionogels with different contents of HP-A were measured. The changes of storage modulus (*G*′) and loss modulus (*G*″) as a function of angular frequency for ionogels are shown in Figure 3a. For all ionogels, the typical rubberlike behavior is observed, as *G*′ is always higher than *G*″, and *G*′ becomes independent of frequency over the entire investigated frequency range. It is also clearly seen from Figure 3a that the storage modulus increases as polymer loading increases, which is due to the fact that network densities increase with increasing polymer concentration. Furthermore, these ionogels have good modulus up to 10^5^ Pa, indicating the appropriate mechanical strengths of these ionogels for applications. Commonly, temperature can negatively impact the performance and stability of electrolytes, especially for the physical cross-linked gel polymer electrolytes [11,14]. However, these chemical cross-linked ionogels display high mechanical stability, and the solid state structure is not destroyed even at temperatures up to 200 °C (Figure 3b). Therefore, the obtained ionogels can be used as high performance electrolytes in high-temperature electronic devices.

Mechanical strength is an important property for gel polymer electrolytes when used and assembled in an electronic device. It can be seen from Figure 3c that the rectangular shape ionogel can be twisted, which demonstrates the ionogel has good mechanical properties. The tensile property tests were performed and the stress–strain curves are shown in Figure 3d. It can be seen that the tensile strength increases with increasing HP-A content, while the elongation at the break decreases with increasing HP-A content. The tensile strength for Ionogel-2 with 15% hyperbanched polymer is 0.3 ± 0.1 MPa, while the tensile strength for Ionogel-5 with 30% hyperbranched polymer sharply increases to 1.6 ± 0.2 MPa and has about a 500% enhancement. These data suggest that the obtained ionogels with high modulus and mechanical strength are desirable as new electrolytes for preparing high performance devices.

### 3.3. Thermal Stability of Ionogel

It is known that the thermal stability of the electrolytes is of great importance for securing device safety. Thermogravimetric analysis (TGA) and differential thermal gravimetric (DTG) curves of ionic liquid BMIMBF_4_, HP-A and ionogels are shown in Figure 4. For ionic liquid BMIMBF_4_, the TGA shows a noticeable change only above 400 °C and the DTG curve shows only a single peak at 457.9 °C, demonstrating the good thermal stability for BMIMBF_4_ as an electrolyte. The hyperbranched polymer of HP-A undergoes a noticeable weight loss process from 385.3 °C, while the DTG curve shows a single peak at 463.5 °C. For the ionogels, the initial thermal decomposition temperature is 371.3 °C, and the DTG curve shows that the tiptop temperature of pyrolysis is 463.5 °C. These results reveal that the obtained ionogels with excellent thermal stability can be highly useful as a gel polymer electrolyte for fabricating high-temperature electrolyte devices.

### 3.4. Ionic Conductivity of Ionogels

The electrochemical properties of these ionogels were measured and the impedance spectra of ionogels with different contents of hyperbranched polymer at room temperature are shown in Figure 5a. The minimum in the Nyquist plot of the negative imaginary part of the impedance versus the real part of the impedance is taken as the sample resistance. It can be seen that the resistance of the ionogels increases with increasing the contents of the hyperbranched polymer. Ionic conductivities can be calculated from the bulk resistances, and the results are shown in Figure 5b. The ionic conductivity of the ionogels decreases with increasing the contents of the hyperbranched polymer, which is due to the decrease of ion concentration. Note that the room temperature ionic conductivity of all the ionogels is still more than 1 mS cm^−1^, even the polymer content is up to 30%. The main reason for this is ascribed to the porous structure with beads-shaped networks and the hyperbranched polymers are completely amorphous with nanocavities, which makes ions move freely. The electrochemical stability of the ionogel was also characterized by linear sweep voltammetry. As shown in Appendix A, the ionogel-5 displays good electrochemical stability and the electrochemical window is 4.3 V, which can meet the demand of the application for lithium-ion batteries.

The Arrhenius plots for the temperature dependence of ionic conductivity of ionic liquid BMIMBF_4_ and ionogels with different contents of hyperbranched polymer over the temperature range of 20–120 °C is also shown in Figure 6. It clearly shows that for all the samples, the ionic conductivity monotonously increases along with increasing temperature. Furthermore, the ionic conductivity of the ionogels decreases with increasing the contents of the hyperbranched polymer. The ionic conductivities of all the ionogels are between 1.2 × 10^−3^ S cm^−1^ and 5.0 × 10^−2^ S cm^−1^, which completely satisfies the requirement for the practical application in Li batteries as gel polymer electrolytes.

### 3.5. Charging/Discharging Performance of Li/LiFePO_4_ Cells

To characterize the electrochemical performance of the obtained ionogels, we fabricated cells using LiFePO_4_ as the cathode and metallic lithium counter electrodes assembled with ionogels (30% HP-A/50% BMIMBF_4_/10% PVDF/10% LiBF_4_). Figure 7a shows that the pouch cell assembled with the obtained ionogels can successfully light up a yellow LED lamp. Typical charge and discharge profiles of the cells at room temperature at a charge-discharge rate of 0.1 C are displayed in Figure 7b. A higher specific capacity of 153.1 mAhg^−1^ is obtained, which is comparable to the literature results [23,38]. The Li/ionogels/LiFePO_4_ cell has been evaluated for cycle ability with a constant charge/discharge current density (0.1 C/0.1 C) at room temperature, and the results are shown in Figure 7c,d. The discharge capacity is maintained at a value of 147.2 mAhg^−1^ after 100 cycles, which is 96.1% of its largest discharge capacity. The columbic efficiency, determined by the ratio between the discharge and charge capacity, is found to be 98.1%, which is very close to 100%, suggesting that the cell exhibits very stable charge/discharge behavior with good cycle performance.

Figure 8a shows the typical charge/discharge profiles of the cells under different temperatures at 0.1 C. It can be seen that the discharge capacity at 80 °C is 201.5 mAhg^−1^, which is much higher than that at 50 °C (179.2 mAhg^−1^) and 25 °C (153.1 mAhg^−1^). Such an improvement in capacity might be related to the increased movement of the polymer chain segment as well as high ion mobility at an elevated temperature. Surprisingly, the values at 50 °C and 80 °C are even in excess of the theoretical value of 170 mAhg^−^^1^ for LiFePO_4_. Similar results have also been observed in lithium ion batteries based on LiFePO_4_ and other anodes, which was mainly ascribed to the special nanostructure of the interface and the reversible redox reaction between the lithium ions and the oxygenic groups (–C=O) at the interface [47,48,49,50]. For our system, the ionogel electrolyte with a special beads-shaped network structure and nanometer-sized pores was wrapped around the cathode material LiFePO_4_, which could cause the interface to have a special nanostructure. Meanwhile, the polymer matrix of the hyperbranched polymer contains lots of oxygenic groups (–C=O). Therefore, a highly-specific capacity beyond theoretical capacity was obtained for the system with special ionogel based on a hyperbranched polymer. The rate performances of the fabricated Li/ionogel/LiFePO_4_ cell are shown in Figure 8b,c. Note that the cell was charged at a constant charge current density of 0.1 C and discharged at various current densities ranging from 0.1 to 1 C. The specific discharge capacities of the cell at 0.1, 0.2, 0.5, and 1 C are 153.1, 141.1, 124.5, and 104.6 mAhg^−1^, respectively (Figure 8b). Interestingly, the original capacity is largely recovered while the C-rate is switched abruptly from 1 C to 0.1 C again (Figure 8c), indicating that the cell is robust and highly stable. As a result, cells with these chemical cross-linked ionogel electrolytes based on a hyperbranched polymer are practical both in the common and high temperature environment.

## 4. Conclusions

In summary, the chemically cross-linked ionogels based on an acrylated hyperbranched polymer and ionic liquid of BMIMBF_4_ were synthesized by photopolymerization. The ionic conductivity of the ionogels decreased with increasing the contents of hyperbranched polymer, but the ionic conductivities of all the ionogels were between 10^−3^ S cm^−1^ and 10^−1^ S cm^−1^, which completely satisfied the requirements for the practical application in Li batteries. The modulus and mechanical strength of the ionogels increased with increasing the contents of the polymer. Importantly, these chemical cross-linked ionogels showed high mechanical stability and the solid state structure was not destroyed even at temperatures up to 200 °C. The obtained ionogels had very high thermal stability, which could sustain 371.3 °C thermal treatments. Moreover, the Li/ionogel/LiFePO_4_ batteries showed a higher specific capacity of 153.1 mAhg^−1^ and exhibited very stable charge/discharge behavior with good cycle performance, for which the capacity could retain 98.1% after 100 cycles. Therefore, the studies suggest that the obtained ionogels based on a hyperbranched polymer are promising candidates for stable and high performance energy storage and conversion devices.

## Figures and Tables

**Figure 1 polymers-11-00444-f001:**
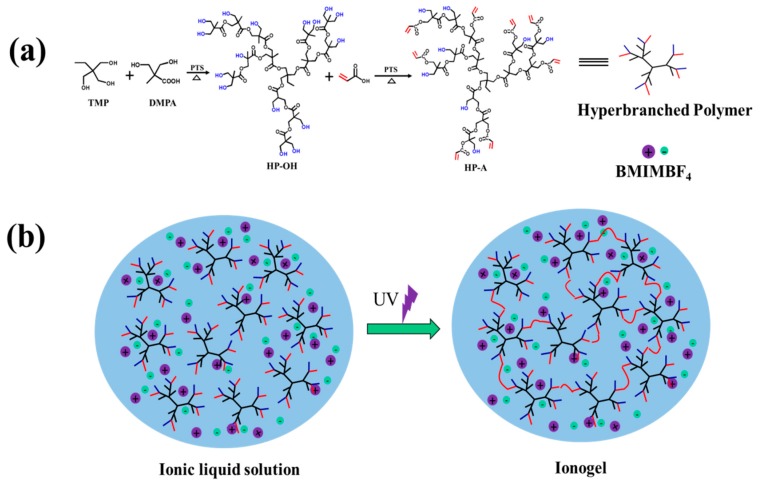
Schematic illustration of (**a**) synthesis of acrylate terminated hyperbranched polymer (HP-A) and (**b**) ionogel from acrylate terminated hyperbranched polymer by photopolymerization.

**Figure 2 polymers-11-00444-f002:**
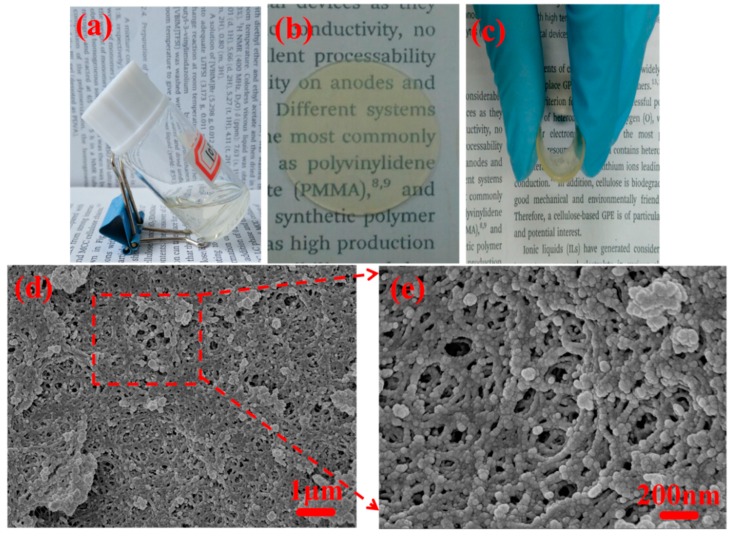
Photographs of (**a**) the ionic liquid solution, (**b**) round shape ionogel, and (**c**) bent round shape ionogel. (**d**,**e**) SEM images ofcross-sections of ionogels.

**Figure 3 polymers-11-00444-f003:**
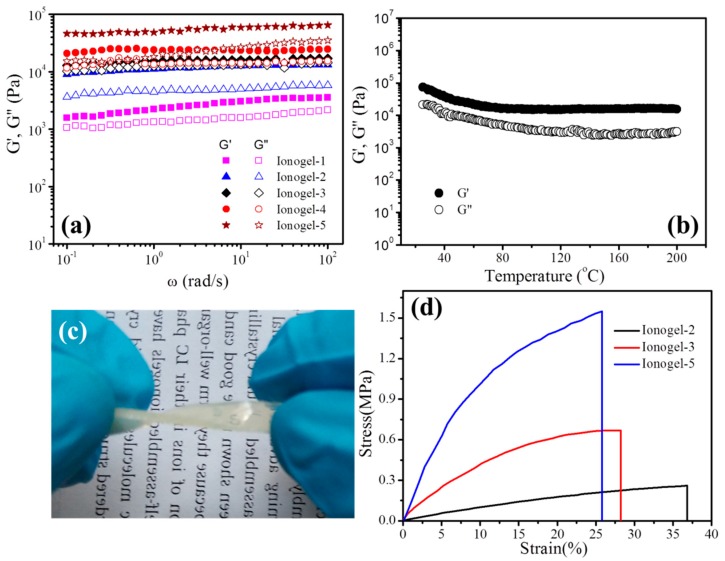
(**a**) Changes of *G*′ and *G*″ as a function of ω at 25 °C for ionogels; (**b**) changes of *G*′ and *G*′ with increasing temperature at an angular frequency of 6.28 rad/s for ionoge-5; (**c**) twisted rectangular shape ionogel; (**d**) stress-strain curves of ionogels with different content of HP-A.

**Figure 4 polymers-11-00444-f004:**
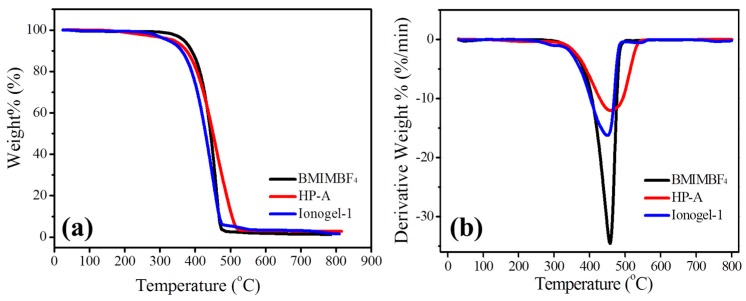
(**a**) TGA and (**b**) DTG curves of BMIMBF_4_, HP-A and ionogels.

**Figure 5 polymers-11-00444-f005:**
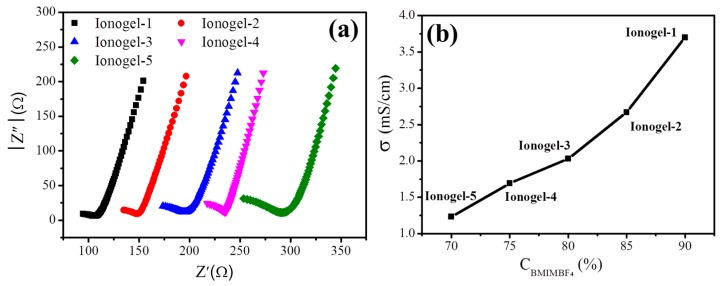
(**a**) The impedance plots of ionogels; (**b**) changes of ionic conductivity as functions of polymer.

**Figure 6 polymers-11-00444-f006:**
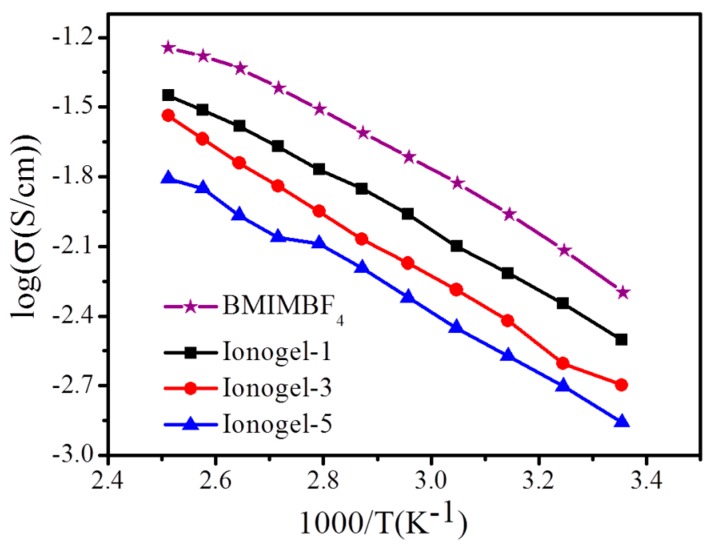
Temperature dependence of ionic conductivities for neat BMIMBF_4_ and ionogel shown in the Arrhenius convention.

**Figure 7 polymers-11-00444-f007:**
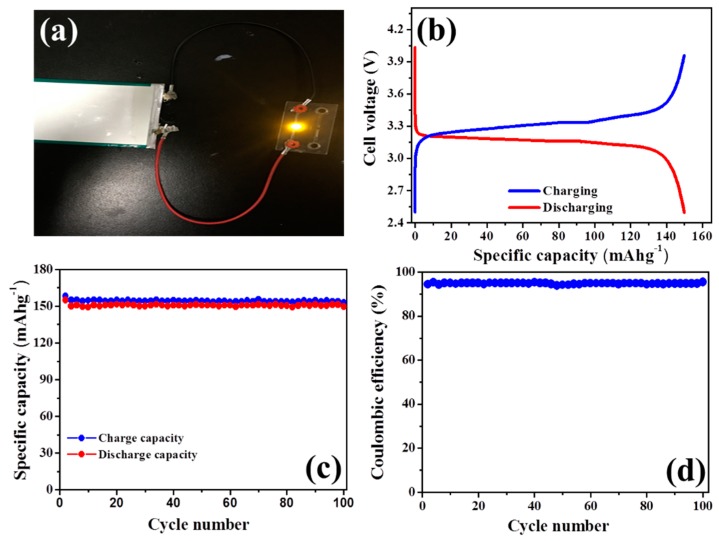
(**a**) A photograph showing a yellow LED lamp (~2.0 V) connected to the cell assembled with the ionogels; (**b**) charge and discharge profiles at room temperature for LiFePO_4_/30% HP-A/50% BMIMBF_4_/10% PVDF/10% LiBF4/Li cells (with a charge-discharge rate of 0.1 C); (**c**) and (**d**) cycling performance of the cell at room temperature for 100 cycles.

**Figure 8 polymers-11-00444-f008:**
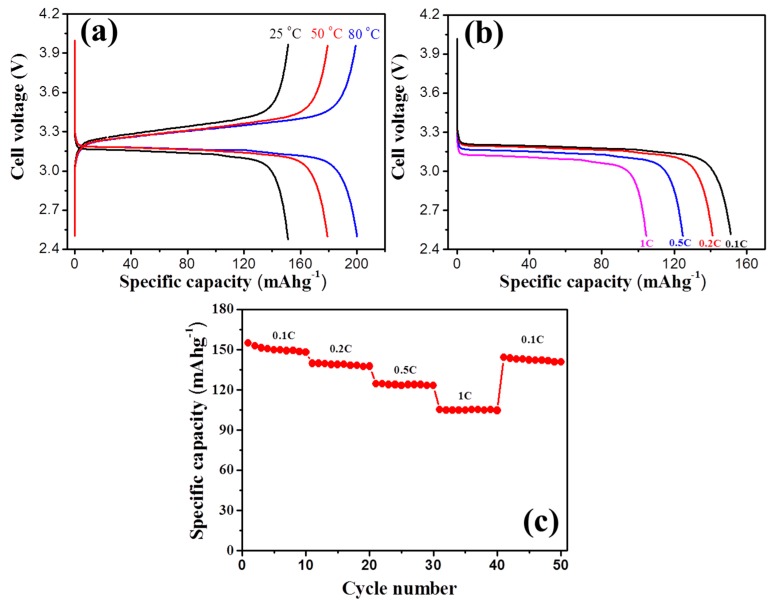
(**a**) Charge and discharge profiles under 25 °C, 50 °C, and 80 °C at a current rate of 0.1 C; (**b**) discharge profiles at various C-rates from 0.1 to 1 C (25 °C); (**c**) rate performances of the cell at 25 °C, where discharge current densities are varied from 0.1 to 1.0 C at a constant charge current density of 0.1 C.

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
