# Peer review of "Novel Chemical Cross-Linked Ionogel Based on Acrylate Terminated Hyperbranched Polymer with Superior Ionic Conductivity for High Performance Lithium-Ion Batteries"

_polymers, 2019, doi:10.3390/polym11030444_

Round 1

Reviewer 1 Report

 Comments to author:

The present manuscript describes the synthesis of ionogel by polymerization in an ionic liquid. Various properties of the obtained microstructures were studied and described along with the LIBs performance. The manuscript is recommended for publication in Polymers if the authors addresses the following issues.

1.      In the Introduction paragraph-2, most of the literature related explanation appears as a hypothesis, for example, lots of, most of, etc. The authors should provide related examples from the citing literature.

2.      It would be better for readers if Schematic illustration (Fig. S1) moved to main text and presented combined with Fig. 1.

3.      Page 5, line- 171. Fig. 4 d, e should be Fig 2 d, e.

4.      The ability of charge and discharge LIBs at higher rate is of enormous importance for practical use. Therefore, authors should provide and explanation the specific capacity measured as a function of current in the range of 0.1-1 C.

Author Response

The present manuscript describes the synthesis of ionogel by polymerization in an ionic liquid. Various properties of the obtained microstructures were studied and described along with the LIBs performance. The manuscript is recommended for publication in Polymers if the authors addresses the following issues.

Response: We sincerely thank Reviewer 1 for her/his nice recommendation.

Comment 1: In the Introduction paragraph-2, most of the literature related explanation appears as a hypothesis, for example, lots of, most of, etc. The authors should provide related examples from the citing literature.

Response: We sincerely thank Reviewer 1 for her/his nice recommendation. We feel sorry for not giving a clear expression in our previous manuscript. The related examples have been shown in the newly revised version. The related content is copied below for your reference.

For example, Lee et al. found that poly(ethylene oxide)s with varying degrees of hyperbranching were effective at preventing the crystallization of PEO and leaded to approximately a 100-fold increase in the Li-ion conductivity below 50 °C as compared with linear PEO.” Line 53 to 56.

“Itoh et al. found that the cross-linked composite polymer electrolytes of cross-linkable hyperbranched polymer capped with acryloyl group showed higher tensile strength than the non-cross-linked composite polymer electrolyte.” Line 58 to 60.

“such as acrylate end groups (the hyperbranched star polymer with hyperbranched polystyrene as core and polymethyl methacrylate block poly(ethylene glycol) methyl ether methacrylate), amino end groups (hyperbranched polyamidoamine).” Line 62 to 64.

Comment 2: It would be better for readers if Schematic illustration (Fig. S1) moved to main text and presented combined with Fig. 1.

Response: We sincerely thank Reviewer 1 for her/his nice recommendation. Taking the advice from the reviewer, the schematic illustration (Fig. S1) was moved to main text and presented combined with Fig. 1 in the in the newly revised version. The related content is copied below for your reference.

Fig.R1 Schematic illustration of (a) synthesis of acrylate terminated hyperbranched polymer (HP-A) and (b) ionogel from acrylate terminated hyperbranched polymer by photopolymerization.

Comment 3: Page 5, line- 171. Fig. 4 d, e should be Fig 2 d, e.

Response: We sincerely thank Reviewer 1 for her/his nice recommendation. We feel sorry for not giving a clear expression in our previous manuscript. This mistake has been corrected carefully in the newly revised version.

Comment 4: The ability of charge and discharge LIBs at higher rate is of enormous importance for practical use. Therefore, authors should provide and explanation the specific capacity measured as a function of current in the range of 0.1-1 C.

Response: We sincerely thank Reviewer 1 for her/his nice recommendation. The date has been shown in the newly revised version. The related content is copied below for your reference.

Fig.R2 (a) discharge profiles at various C-rates from 0.1 to 1 C (25 oC); (b) rate performances of the cell at 25 oC, where discharge current densities are varied from 0.1 to 1.0 C at a constant charge current density of 0.1 C.

“The rate performances of the fabricated Li/ionogel/LiFePO4 cell are shown in Fig. 8b and c. Note that the cell was charged at a constant charge current density of 0.1 C and discharged at various current densities ranging from 0.1 to 1 C. The specific discharge capacities of the cell at 0.1, 0.2, 0.5 and 1 C are 153.1, 141.1, 124.5 and 104.6 mAhg−1, respectively (Fig. 8b). Interestingly, the original capacity is largely recovered while the C-rate is switched abruptly from 1C to 0.1C again (Fig. 8c), indicating that the cell is robust and highly stable. As a result, cells with these chemical cross-linked ionogels electrolytes based on hyperbranched polymer are practical both in the common and high temperature environment.”

Reviewer 2 Report

Manuscript is well written and claims are supported by experimental data. Following points need to be clarified/answered before the final acceptance:

1. How come the observed capacity is exceeding the theoretical capacity of LiFePO4 in figure 8(a)?  It is mentioned that capacity enhanced due to increased movement of polymer chain, but this concept doesn't explain the reason as there are limited number of Li ions to intercalate/deinterclaalte. Capacity depends upon number of Li ions transferred. Actual capacity cannot exceed the theoretical unless until there's some othe electrochemical process is involved. 

Author Response

Manuscript is well written and claims are supported by experimental data. Following points need to be clarified/answered before the final acceptance:

Response: We sincerely thank Reviewer 2 for her/his nice recommendation.

Comment 1: How come the observed capacity is exceeding the theoretical capacity of LiFePO4 in figure 8(a)?  It is mentioned that capacity enhanced due to increased movement of polymer chain, but this concept doesn't explain the reason as there are limited number of Li ions to intercalate/deinterclaalte. Capacity depends upon number of Li ions transferred. Actual capacity cannot exceed the theoretical unless until there's some other electrochemical process is involved.

Response: We sincerely thank Reviewer 1 for her/his nice recommendation.

Commonly, actual capacity of LiFePO4 cannot exceed the theoretical 170 mAhg-1. The reason is that some lithium ions cannot be fully extracted from the ordered-olivine structure due to the low ionic mobility, which in turn causes some capacity loss. Enormous efforts have been made to conquer the drawback and some reports got very high value that beyond the theoretical (208 mAhg-1 in paper “Nature communications 2013, 4, 1687-1687”; 192 mAhg-1 in paper “Nano Energy 2017, 34, 408-420.”  Similar results were also observed in lithium ion batteries based on SnO2 anode (Scientific Reports 2015, 5, 9164) and Cu-based integrated anode (Journal of Power Sources 2015, 275, 136-143.). It is speculated that the extra capacity is mainly ascribed to the special nanostructure of the interface and the reversible redox reaction between the lithium ions and the oxygenic groups (-C=O) at the interface. For our system, the ionogel electrolyte that with special beads-shaped network structure and nanometer-sized pores was wrapped around the cathode material LiFePO4, which could make the interface having special nanostructure. Meanwhile, polymer matrix of hyperbranched polymer contains lots of oxygenic groups (-C=O). Therefore, high specific capacity that beyond theoretical capacity was obtained for the system with special ionogel based on hyperbranched polymer.

We feel sorry for not giving a clear expression in our previous manuscript. The related discussion is presented as follows:

“Such improvement in capacity might be related to the increased movement of the polymer chain segment as well as high ion mobility at elevated temperature. Surprisingly, the values at 50 oC and 80 oC are even in excess of the theoretical value of 170 mAhg-1 for LiFePO4. Similar results have also been observed in lithium ion batteries based on LiFePO4 and other anodes, which was mainly ascribed to the special nanostructure of the interface and the reversible redox reaction between the lithium ions and the oxygenic groups (-C=O) at the interface. For our system, the ionogel electrolyte that with special beads-shaped network structure and nanometer-sized pores was wrapped around the cathode material LiFePO4, which could make the interface having special nanostructure. Meanwhile, polymer matrix of hyperbranched polymer contains lots of oxygenic groups (-C=O). Therefore, high specific capacity that beyond theoretical capacity was obtained for the system with special ionogel based on hyperbranched polymer.”

Related references:

[1]Zhao, Q.; Zhang, Y.; Meng, Y.; Wang, Y.; Ou, J.; Guo, Y.; Xiao, D., Phytic acid derived LiFePO4 beyond theoretical capacity as high-energy density cathode for lithium ion battery. Nano Energy 2017, 34, 408-420.

[2] Hu, L.; Wu, F.-Y.; Lin, C.-T.; Khlobystov, A. N.; Li, L.-J., Graphene-modified LiFePO cathode for lithium ion battery beyond theoretical capacity. Nature communications 2013, 4, 1687-1687.

[3]Wang, Y.; Huang, Z. X.; Shi, Y. M.; Wong, J. I.; Ding, M.; Yang, H. Y., Designed hybrid nanostructure with catalytic effect: beyond the theoretical capacity of SnO2 anode material for lithium ion batteries. Scientific Reports 2015, 5, 9164.

[4]Chen, K.; Xue, D.; Komarneni, S., Beyond theoretical capacity in Cu-based integrated anode: Insight into the structural evolution of CuO. Journal of Power Sources 2015, 275, 136-143.

Reviewer 3 Report

1. How much initiator was used?  Currently it is stated in line 100, "To this HP-A/IL mixture, amount of 2-Hydroxy-4’-(2-hydroxyethoxy)-2-methylpropiophenone (as the initiator) was added and stirred..."

2. Figure 1b appears to be a misleading depiction of the network structure.  The figure shows that the HP-A units shown in (a) are connected by linear chains upon crosslinking.  But HP-A is the only monomer listed in the reaction solution, therefore the multifunctional HP-A nodes will link directly together.

3. In Figure 8a caption, the charge/discharge rate is not listed.  

4. This manuscript appears thorough.  I note that the properties obtained here for this ionic liquid gel polymer electrolyte are quite similar to those obtained for other ionic liquid gel polymer electrolytes reported in the literature, despite the fancy nature of the crosslinked polymer.  

Formatting / typographical errors:

- Line 74: "Acrylate terminated hyperbranched polymers were synthesized viatwo-step process" should be "Acrylate terminated hyperbranched polymers were synthesized via a two-step process"

- In many places, there is a space missing between the number and associated unit. Ex., "1.34g" rather than "1.34 g"

-Line 233: please fix "is still more than 1mS cm-1 evens the polymer content up to 30%."

Author Response

Comments to the Author

Comment 1: How much initiator was used?  Currently it is stated in line 100, "To this HP-A/IL mixture, amount of 2-Hydroxy-4’-(2-hydroxyethoxy)-2-methylpropiophenone (as the initiator) was added and stirred..."

Response: We sincerely thank Reviewer 3 for her/his nice recommendation. We feel sorry for not giving a clear expression in our previous manuscript. In our paper, 1 wt% (mass fraction with respect to HP-A content) of 2-Hydroxy-4’-(2-hydroxyethoxy)-2-methylpropiophenone (as the initiator) was added.

Comment 2: Figure 1b appears to be a misleading depiction of the network structure.  The figure shows that the HP-A units shown in (a) are connected by linear chains upon crosslinking.  But HP-A is the only monomer listed in the reaction solution, therefore the multifunctional HP-A nodes will link directly together.

Response: We sincerely thank Reviewer 3 for her/his nice recommendation. We feel sorry for not giving a clear expression in our previous manuscript. The picture has been checked and corrected carefully. The date has been shown in the newly revised version. The related content is copied below for your reference.

Fig.R1 Schematic illustration of (a) synthesis of acrylate terminated hyperbranched polymer (HP-A) and (b) ionogel from acrylate terminated hyperbranched polymer by photopolymerization.

Comment 3: In Figure 8a caption, the charge/discharge rate is not listed. 

Response: We sincerely thank Reviewer 3 for her/his nice recommendation. We feel sorry for not giving a clear expression in our previous manuscript. The charge/discharge rate is 0.1C and this mistake has been corrected carefully in the newly revised version.

“Fig. 8 (a) Charge and discharge profiles under 25 oC, 50 oC, and 80 oC at current rate of 0.1 C.”

Comment 4: This manuscript appears thorough.  I note that the properties obtained here for this ionic liquid gel polymer electrolyte are quite similar to those obtained for other ionic liquid gel polymer electrolytes reported in the literature, despite the fancy nature of the crosslinked polymer. 

Response: We sincerely thank Reviewer 3 for her/his nice recommendation.

      In our paper, the ionic conductivities of all the ionogels based on hyperbranched polymers are at between 10-3 S cm-1 and 10-1 S cm-1, which higher than some other ionic liquid gel polymer electrolytes (ionic conductivities are 10-4 -10-2 S cm-1) at the same amount of polymer. This is because the hyperbranched polymers posess completely amorphous and show low glass transition temperatures. Meawhile, the mechanical strength of these ionogels (105-106 Pa) is also higher than than some other ionic liquid gel polymer electrolytes (104-106 Pa), due to that he hyperbranched polymers have more functional groups and the ionogels have more network densities.

Comment 5:.Formatting / typographical errors:

- Line 74: "Acrylate terminated hyperbranched polymers were synthesized viatwo-step process" should be "Acrylate terminated hyperbranched polymers were synthesized via a two-step process"

- In many places, there is a space missing between the number and associated unit. Ex., "1.34g" rather than "1.34 g"

-Line 233: please fix "is still more than 1mS cm-1 evens the polymer content up to 30%."

Response: We sincerely thank Reviewer 1 for her/his nice recommendation. We feel sorry for not giving a clear expression in our previous manuscript. These mistakes have been corrected carefully in the newly revised version. The related content is copied below for your reference.

- Line 74: "Acrylate terminated hyperbranched polymers were synthesized viatwo-step process" has been changed into "Acrylate terminated hyperbranched polymers were synthesized via a two-step process".

All space missing between the number and associated unit were added.

The sentence "is still more than 1 mS cm-1 evens the polymer content up to 30%." has been changed into "is still more than 1 mS cm-1 even the polymer content up to 30%".

Round 2

Reviewer 2 Report

Manuscript can be accepted based on the references provided and added explanations.